# UNIFYING QUESTION ANSWERING, TEXT CLASSIFICATION, AND REGRESSION VIA SPAN EXTRACTION

## ABSTRACT

Even as pre-trained language encoders such as BERT are shared across many tasks, the output layers of question answering, text classification, and regression models are significantly different. Span decoders are frequently used for question answering, fixed-class, classification layers for text classification, and similarity-scoring layers for regression tasks, We show that this distinction is not necessary and that all three can be unified as span extraction. A unified, span-extraction approach leads to superior or comparable performance in supplementary supervised pre-trained, low-data, and multi-task learning experiments on several question answering, text classification, and regression benchmarks.

## 1 INTRODUCTION

Pre-trained natural language processing (NLP) systems (Radford et al., 2019; Devlin et al., 2018; Radford et al., 2018; Howard & Ruder, 2018; Peters et al., 2018; McCann et al., 2017; Liu et al., 2019b) have been shown to transfer remarkably well on downstream tasks including text classification, question answering, machine translation, and summarization (Wang et al., 2018; Rajpurkar et al., 2016; Conneau et al., 2018). Such approaches involve a pre-training phase followed by the addition of task-specific layers and a subsequent re-training or fine-tuning of the conjoined model. Each task-specific layer relies on an inductive bias related to the kind of target task. For question answering, a task-specific span-decoder is often used to extract a span of text verbatim from a portion of the input text (Xiong et al., 2016). For text classification, a task-specific classification layer with fixed classes is typically used instead. For regression, similarity-measuring layers such as least-squares and cosine similarity are employed. These task-specific inductive biases are unnecessary. On several tasks predominantly treated as text classification or regression, we find that reformulating them as span-extraction problems and relying on a span-decoder yields superior performance to using a task-specific layers.

For text classification and regression problems, pre-trained NLP systems can benefit from supplementary training on intermediate-labeled tasks (STILTs) (Phang et al., 2018), i.e. supplementary supervised training. We find this is similarly true for question answering, classification, and regression when reformulated as span-extraction. Because we rely only on the span-extractive inductive bias, we are able to further explore previously unconsidered combinations datasets. By doing this, we find that question answering tasks can benefit from text classification tasks and classification tasks can benefit from question answering ones.

The success of pre-training for natural language processing systems affords the opportunity to re-examine the benefits of our inductive biases. Our results on common question answering, text classification, and regression benchmark tasks suggest that it is advantageous to discard the inductive bias that motivates task-specific, fixed-class, classification and similarity-scoring layers in favor of the inductive bias that views all three as span-extraction problems.

### 1.1 CONTRIBUTIONS

Summarily, we demonstrate the following:

1. Span-extraction is an effective approach for unifying question answering, text classification, and regression.

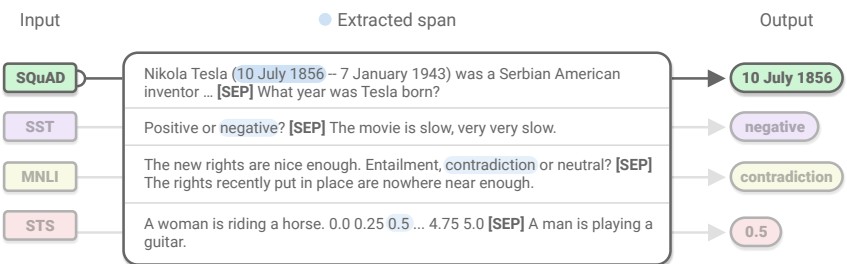

Figure 1: Illustration of our proposed approach using the BERT pre-trained sentence encoder. Text classification tasks are posed as those of span extraction by appending the choices to the input. Similarly, regression tasks are posed by appending bucketed values to the input. For question answering, no changes over the BERT approach are necessary. The figure includes four examples from the SQuAD, SST, MNLI, and STS datasets, respectively.

2. Span-extraction benefits as much from intermediate-task training as more traditional text classification and regression methods.

3. Span-extraction allows for combinations of question answering and text classification datasets in intermediate-task training that outperform using only one or the other.

4. Span-extractive multi-task learning yield stronger multi-task models, but weaker single-task models compared to intermediate-task training.

5. Span-extraction with intermediate-task training proves more robust in the presence of limited training data than the corresponding task-specific versions.

## 2 RELATED WORK

**Transfer Learning.** The use of pre-trained encoders for transfer learning in NLP dates back to Collobert & Weston (2008); Collobert et al. (2011) but has had a resurgence in the recent past. BERT (Devlin et al., 2018) employs the recently proposed Transformer layers (Vaswani et al., 2017) in conjunction with a masked language modeling objective as a pre-trained sentence encoder. Prior to BERT, contextualized word vectors (McCann et al., 2017) were pre-trained using machine translation data and transferred to text classification and question answering tasks. ELMO (Peters et al., 2018) improved contextualized word vectors by using a language modeling objective instead of machine translation. ULMFit (Howard & Ruder, 2018) and GPT (Radford et al., 2018) showed how traditional, causal language models could be fine-tuned directly for a specific task, and GPT-2 (Radford et al., 2019) showed that such language models can indirectly learn tasks like machine translation, question answering, and summarization.

**Intermediate-task and Multi-task Learning.** The goal of unifying NLP is not new (Collobert & Weston, 2008; Collobert et al., 2011). In Phang et al. (2018), the authors explore the efficacy of supplementary training on intermediate tasks, a framework that the authors abbreviate as STILTs. Given a target task $T$ and a pre-trained sentence encoder, they first fine-tune the encoder on an intermediate (preferably related) task $I$ and then finally fine-tune on the task $T$. The authors showed that such an approach has several benefits including improved performance and better robustness to hyperparameters. The authors primarily focus on the GLUE benchmark (Wang et al., 2018). Liu et al. (2019a) explore the same task and model class (viz., BERT) in the context of multi-tasking. Instead of using supplementary training, the authors choose to multi-task on the objectives and, similar to BERT on STILTs, fine-tune on the specific datasets in the second phase. Further improvements can be obtained through heuristics such as knowledge distillation as demonstrated in Clark et al. (2019). All of these approaches require a different classifier head for each task, e.g., a two-way classifier for SST and a three-way classifier for MNLI. Two recent approaches: decaNLP (McCann et al., 2018) and GPT-2 Radford et al. (2019) propose the unification of NLP as question answering and language modeling, respectively. As investigated in this work, the task description is provided in natural language instead of fixing the classifier a-priori.

| Task | Dataset | Source Text | Auxiliary Text |
|---|---|---|---|
| Sentence Classification | SST | positive or **negative**? | it's slow – very, very slow |
| Sentence Pair Classification | MNLI | I don't know a lot about camping. entailment, **contradiction**, or neutral? | I know exactly. |
| Sentence Pair Classification | RTE | The capital of Slovenia is Ljubljana, with 270,000 inhabitants. entailment or **not**? | Slovenia has 270,000 inhabitants. |
| Sentence Pair Regression | STS-B | A woman is riding a horse. 0.0 0.25 **0.5** 0.75 1.0 $\cdots$ 5.0. | A man is playing a guitar. |
| Question Answering | SQuAD | Nikola Tesla (**10 July 1856** – 7 January 1943) was a Serbian American inventor ... | When was Tesla born? |

Table 1: Treating different examples as forms of span-extraction problems. For sentence pair classification datasets, one sentence is present in each of the source text and auxiliary text. The possible classification labels are appended to the source text. For single sentence classification datasets, the source text only contains the possible classification labels. For question answering datasets, no changes to the BERT formulation is required; the context is presented as source text and the question as auxiliary text.

## 3 METHODS

We propose treating question answering, text classification, and regression as span-extractive tasks. Each input is split into two segments: a source text which contains the span to be extracted and an auxiliary text that is used to guide extraction. Question answering often fits naturally into this framework by providing both a question and a context document that contains the answer to that question. When treated as span-extraction, the question is the auxiliary text and the context document is the source text from which the span is extracted. Text classification input text most often does not contain a natural language description of the correct class. When it is more natural to consider the input text as one whole, we treat it as the auxiliary text and use a list of natural language descriptions of all possible classification labels as source text. When the input text contains two clearly delimited segments, one is treated as auxiliary text and the other as source text with appended natural language descriptions of possible classification labels. For regression, we employ a process similar to classification; instead of predicting a floating-point number, we bucket the possible range and classify the text instead.

Our proposal is agnostic to the details of most common preprocessing and tokenization schemes for the tasks under consideration, so for ease of exposition we assume three phases: preprocessing, encoding, and decoding. Preprocessing includes any manipulation of raw input text; this includes tokenization. An encoder is used to extract features from the input text, and an output layer is used to decode the output from the extracted features. Encoders often include a conversion of tokens to distributed representation followed by application of several layers of LSTM, Transformer, convolutional neural network, attention, or pooling operations. In order to properly use these extracted features, the output layers often contain more inductive bias related to the specific task. For many question answering tasks, a span-decoder uses the extracted features to select a start and end token in the source document. For text classification, a linear layer and softmax allow for classification of the extracted features. Similarly, for regression, a linear layer and a similarity-scoring objective such as cosine distance or least-squares is employed. We propose to use span-decoders as the output layers for text classification and regression in place of the more standard combination of linear layer with task-specific objectives.

### 3.1 SPAN-EXTRACTIVE BERT (SPEX-BERT)

In our experiments, we start with a pre-trained BERT as the encoder with preprocessing as described in Devlin et al. (2018). This preprocessing takes in the source text and auxiliary text and outputs a sequence of $p = m + n + 2$ tokens: a special CLS token, the $m$ tokens of the source text, a separator

token SEP, and the $n$ auxiliary tokens. The encoder begins by converting this sequence of tokens into a sequence of $p$ vectors in $\mathbb{R}^d$. Each of these vectors is the sum of a token embedding, a positional embedding that represents the position of the token in the sequence, and a segment embedding that represents whether the token is in the source text or the auxiliary text as described in Devlin et al. (2018). This sequence is stacked into a matrix $X_0 \in \mathbb{R}^{p \times d}$ so that it can be processed by several Transformer layers (Vaswani et al., 2017). The $i$th layer first computes $\alpha^p(X_i)$ by applying self-attention with $k$ heads over the previous layer's outputs:

$$\alpha^k(X_i) = [h_1; \cdots ; h_k]W_o \tag{1}$$

$$\text{where } h_j = \alpha(X_i W_j^1, X_i W_j^2, X_i W_j^3)$$

$$\alpha(X, Y, Z) = \text{softmax}\left(\frac{XY^\top}{\sqrt{d}}\right) Z \tag{2}$$

A residual connection (He et al., 2016) and layer normalization (Ba et al., 2016) merge information from the input and the multi-head attention:

$$H_i = \text{LayerNorm}(\alpha^p(X_i) + X_i) \tag{3}$$

This is followed by a feedforward network with ReLU activation (Nair & Hinton, 2010; Vaswani et al., 2017), another residual connection, and a final layer normalization. With parameters $U \in \mathbb{R}^{d \times f}$ and $V \in \mathbb{R}^{f \times d}$:

$$X_{i+1} = \text{LayerNorm}(\max(0, H_i U)V + H_i)) \tag{4}$$

Let $X_{sf} \in \mathbb{R}^{m \times d}$ represent the final output of these Transformer layers. At this point, a task-specific head usually uses some part of $X_{sf}$ to classify, regress, or extract spans. Our proposal is to use a span-decoder limited to $X_{sf}$ whenever a classification or similarity-scoring layer is typically used. In this case, we add only two trainable parameter vectors $d_{start}$ and $d_{end}$ following Devlin et al. (2018), and we compute start and end distributions over possible spans by multiplying these vectors with $H_f$ and applying a softmax function:

$$p_{start} = \text{softmax}(X_{sf} d_{start}) \qquad\qquad p_{end} = \text{softmax}(X_{sf} d_{end}) \tag{5}$$

During training, we are given the ground truth answer span $(a^*, b^*)$ as a pair of indices into the source text. The summation of cross-entropy losses over the start and end distributions then gives an overall loss for a training example:

$$\mathcal{L}_{start} = -\sum_i \text{I}\{a^* = i\} \log p_{start}(i) \qquad \mathcal{L}_{end} = -\sum_i \text{I}\{b^* = i\} \log p_{end}(i) \tag{6}$$

With $\mathcal{L} = \mathcal{L}_{start} + \mathcal{L}_{end}$ and at inference, we extract a span $(a, b)$ as

$$a = \arg\max_i p_{start}(i) \qquad\qquad b = \arg\max_i p_{end}(i) \tag{7}$$

## 4 EXPERIMENTAL SETUP

### 4.1 TASKS, DATASETS AND METRICS

We divide our experiments into three categories: classification, regression, and question answering. For classification and regression, we evaluate on all the GLUE tasks (Wang et al., 2018). This includes the Stanford Sentiment Treebank (SST) (Socher et al., 2013), MSR Paraphrase Corpus (MRPC) (Dolan & Brockett, 2005), Quora Question Pairs (QQP), Multi-genre Natural Language Inference (MNLI) (Williams et al., 2017), Recognizing Textual Entailment (RTE) (Dagan et al., 2010; Bar-Haim et al., 2006; Giampiccolo et al., 2007; Bentivogli et al., 2009), Question-answering as NLI (QNLI) (Rajpurkar et al., 2016), and Semantic Textual Similarity (STS-B) Cer et al. (2017). The Winograd schemas challenge as NLI (WNLI) Levesque et al. (2012) was excluded during training because of known issues with the dataset. As with most other models on the GLUE leaderboard, we report the majority class label for all instances. With the exception of STS-B, which is a regression dataset, all other datasets are classification datasets. For question answering, we employ 6 popular

| # Train Ex. | SST 67k | MRPC 3.7k | QQP 364k | MNLI 393k | RTE 2.5k | QNLI 105k | CoLA 8.5k | STS 7k | GLUE Leaderboard Score |
|---|---|---|---|---|---|---|---|---|---|
| *Development Set Scores* | | | | | | | | | |
| BERT$_{LARGE}$ | 92.5 | 89.0 | **91.5** | 86.2 | 70.0 | **92.3** | 62.1 | 90.2 | — |
| →MNLI | 93.2 | 89.5 | 91.4 | 86.2 | 83.4 | **92.3** | 59.8 | **90.9** | — |
| →SNLI | 92.7 | 88.5 | 90.8 | 86.1 | 80.1 | — | 57.0 | 90.7 | — |
| SpEx-BERT$_{LARGE}$ | 93.7 | 88.9 | 91.0 | **86.4** | 69.8 | 91.8 | **64.8** | 89.5 | — |
| →SQuAD | 93.7 | 86.5 | 90.9 | 86.0 | 74.7 | 91.8 | 57.8 | 90.1 | — |
| →TriviaQA (Web) | 93.3 | 85.0 | 90.5 | 85.7 | 73.6 | 91.7 | 60.2 | 89.9 | — |
| →TriviaQA (Wiki) | **94.4** | 86.5 | 90.6 | 85.6 | 71.5 | 91.6 | 59.9 | 90.1 | — |
| →MNLI | **94.4** | **90.4** | 91.3 | **86.4** | **85.2** | 92.0 | 60.6 | **90.9** | — |
| →MNLI→SQuAD | 93.7 | 89.5 | 91.1 | 86.4 | 84.1 | **92.3** | 60.5 | 90.2 | — |
| *Test Set Scores (both on STILTs)* | | | | | | | | | |
| BERT$_{LARGE}$ | 94.3 | 86.6 | 89.4 | 86.0 | **80.1** | **92.7** | 62.1 | 88.5 | 82.0 |
| SpEx-BERT$_{LARGE}$ | **94.5** | **87.6** | **89.5** | **86.2** | 79.8 | 92.4 | **63.2** | **89.3** | **82.3** |

Table 2: Performance metrics on the GLUE tasks. We use Matthew's correlation for CoLA, an average of the Pearson and Spearman correlation for STS, and exact match accuracy for all others. **Bold** marks the best performance for a task in a section delimited by double horizontal lines. Scores for MNLI are averages of matched and mismatched scores. ($\rightarrow A$) indicates that a model was fine-tuned on $A$ as an intermediate task before fine-tuning on a target task (the task header for any particular column). In cases where $A$ and the target task are the same, no additional fine-tuning is done. The phrase *on STILTs* indicates that test set scores on the target task are the result of testing with the best ($\rightarrow A$) according to development scores.

datasets: the Stanford Question Answering Dataset (SQuAD) (Rajpurkar et al., 2016), QA Zero-shot Relationship Extraction (ZRE; we use the $0^{th}$ split and append the token `unanswerable` to all examples so it can be extracted as a span) (Levy et al., 2017), QA Semantic Role Labeling (SRL) (He et al., 2015), Commonsense Question Answering (CQA; we use version 1.0) (Talmor et al., 2018) and the two versions (Web and Wiki) of TriviaQA (Joshi et al., 2017). Unless specified otherwise, all scores are on development sets. Concrete examples for several datasets can be found in Table 1.

## 4.2 TRAINING DETAILS

For training the models, we closely follow the original BERT setup Devlin et al. (2018) and Phang et al. (2018). We refer to the 12-layer model as BERT$_{BASE}$ and the 24-layer model as BERT$_{LARGE}$. Unless otherwise specified, we train all models with a batch size of 20 for 5 epochs. For the SQuAD and QQP datasets, we train for 2 epochs. We coarsely tune the learning rate but beyond this, do not carry out any significant hyperparameter tuning. For STILTs experiments, we re-initialize the Adam optimizer with the introduction of each intermediate task. For smaller datasets, BERT (especially BERT$_{LARGE}$) is known to exhibit high variance across random initializations. In these cases, we repeat the experiment 20 times and report the best score as is common in prior work (Phang et al., 2018; Devlin et al., 2018). The model architecture, including the final layers, stay the same across all tasks and datasets – no task-specific classifier heads or adaptations are necessary.

## 4.3 MODELS AND CODE

Pre-trained models and code can be found at `MASKED`. We rely on the BERT training library[1] available in PyTorch Paszke et al. (2017).

## 5 RESULTS

Next, we present numerical experiments to buttress the claims presented in Section 1.1.

---

[1]`https://github.com/huggingface/pytorch-pretrained-BERT/`

| # Training Examples | SQuAD 87.6k | ZRE 840k | SRL 6.4k | CQA 9.5k |
|---|---|---|---|---|
| SpEx-BERT$_{LARGE}$ | 84.0 | 69.1 | 90.3 | 60.3 |
| → MNLI | **84.5** | 71.6 | 90.7 | 56.7 |
| → ZRE | 84.0 | 69.1 | 90.8 | 61.3 |
| → SQuAD | 84.0 | **82.5** | **91.7** | 63.8 |
| → TriviaQA (Web) | **84.5** | 75.3 | 91.3 | 63.8 |
| → TriviaQA (Wiki) | 84.3 | 74.2 | 91.4 | 64.4 |
| → MNLI → SQuAD | 84.5 | 80.1 | 91.5 | **65.7** |

(a) Exact match scores on the development set for a set of question answering tasks. **Bold** marks the best performance for a task. Note that SpEx-BERT and BERT are equivalent for the question answering task.

| | SST | MRPC | RTE |
|---|---|---|---|
| At most 1k training examples | | | |
| BERT$_{LARGE}$ | 91.1 | 83.8 | 69.0 |
| →MNLI | 90.5 | 85.5 | **82.7** |
| SpEx-BERT$_{LARGE}$ | **91.3** | 82.5 | 67.1 |
| →MNLI | 91.2 | **86.5** | **82.7** |

(b) Development set accuracy scores on three of the GLUE tasks when fine-tuned only on a constrained subset of examples. **Bold** indicates best score for a task.

Table 3

**Span-extraction is similar or superior to task-specific heads (classification or regression).** Table 2 shows our results comparing BERT (with and without STILTs) with the corresponding variant of SpEx-BERT on the GLUE tasks Wang et al. (2018). For almost all datasets, the performance for SpEx-BERT is better than that of BERT, which is perhaps especially surprising for the regression task (STS-B). One can reasonably expect model performance to improve by converting such problems into a span-extraction problem over natural language class descriptions.

**SpEx-BERT improves on STILTs.** As in the case of Phang et al. (2018), we find that using supplementary tasks for pre-training improves the performance on the target tasks. We follow the setup of Phang et al. (2018) and carry out a two-stage training process. First, we fine-tune the BERT model with a span-extraction head on an intermediate task. Next, we fine-tune this model on the target task with a fresh instance of the optimizer. Note that Phang et al. (2018) require a new classifier head when switching between tasks that have different numbers of classes or task, but no such modifications are necessary when SpEx-BERT is applied. SpEx-BERT also allows for seamless switching between question answering, text classification, and regression tasks.

In Table 5, we present the results for SpEx-BERT on STILTs. In a majority of cases, the performance of SpEx-BERT matches or outperforms that of BERT. This is especially pronounced for datasets with limited training data, such as MRPC and RTE with SpEx-BERT$_{LARGE}$ and BERT$_{LARGE}$: 85.2 vs 83.4 for RTE, and 90.4 vs 89.5 for MRPC). We hypothesize that this increase is due to the fact that the class choices are provided to the model *in natural language*, which better utilizes the pre-trained representations of a large language model like BERT. Finally, we note, perhaps surprisingly, that question answering datasets (SQuAD and TriviaQA) improve performance of some of the classification tasks. Notable examples include SST (pre-trained from the Wiki version of TriviaQA) and RTE (pre-trained from any of the three datasets).

**STILTs improves question answering as well.** Table 3a shows similar experiments on popular question answering datasets. The transferability of question answering datasets is well-known. Datasets such as TriviaQA, SQuAD and ZRE have been known to improve each other's scores and have improved robustness to certain kinds of queries (Devlin et al., 2018; McCann et al., 2018). We further discover that through the formulation of SpEx-BERT, classification datasets also help question answering datasets. In particular, MNLI improves the scores of almost all datasets over their baselines. For SQuAD, the benefit of STILTs with the classification dataset MNLI is almost as much as the question answering dataset TriviaQA.

**STILTs can be chained.** Pre-training models using intermediate tasks with labeled data has been shown to be useful in improving performance. Phang et al. (2018) explored the possibility of using one intermediate task to demonstrate this improvement. We explore the possibility of chaining multiple intermediate tasks in Table 3a. Conceptually, if improved performance on SQuAD during the first stage of fine-tuning leads to improved performance for the target task of CQA, improving performance of SQuAD through in turn pre-training it on MNLI would improve the eventual goal of CQA. Indeed, our experiments suggest the efficacy of chaining intermediate tasks in this way. CQA

| Model | RTE |
|---|---|
| $\text{BERT}_{\text{LARGE}} \rightarrow$ RTE | 70.0 |
| $\text{BERT}_{\text{LARGE}} \rightarrow$ MNLI $\rightarrow$ RTE | 83.4 |
| $\text{SpEx-BERT}_{\text{LARGE}} \rightarrow$ RTE | 69.8 |
| $\text{SpEx-BERT}_{\text{LARGE}} \rightarrow$ MNLI $\rightarrow$ RTE | **85.2** |
| $\text{SpEx-BERT}_{\text{LARGE}} \rightarrow \{$MNLI, RTE$\}$ | 75.0 |
| $\text{SpEx-BERT}_{\text{LARGE}} \rightarrow \{$MNLI, RTE$\} \rightarrow$ RTE | 75.8 |

| Natural language description | MNLI |
|---|---|
| Proposed Approach | 84.7 |
| - segmentation of input text | 83.2 |
| - terse class descriptions | 84.4 |

(a) Development set accuracy on the RTE dataset with STILTs and multi-tasking. We denote the process of multi-tasking on datasets $A$ and $B$ by $\{A, B\}$. For each progression (represented by $\rightarrow$), we reset the optimizer but retain model weights from the previous stage.

(b) Development set accuracy using the SpEx-BERT approach on three versions of the MNLI dataset: (1) with the hypothesis and premise separated across source and auxiliary text (see Section 3 for details) and terse class descriptions; (2) with both hypothesis and premise treated entirely as auxiliary text; and (3) with segmented input but including a one-sentence description of the classes (entailment, contradiction, neutral) based on definitions and common synonyms.

Table 4

obtains a score of 63.8 when fine-tuned from a SQuAD model (of score 84.0) and obtains a score of 65.7 when fine-tuned on a SQuAD model that was itself fine-tuned using MNLI (of score 84.5) as an intermediate task.

**Multi-task STILTs yields stronger multi-task models, but weaker single-task models.** We also experiment with multi-task learning during intermediate-task training. We present the results for such intermediate-multi-task training on RTE in Table 4a. In intermediate-multi-task training, we cycle through one batch for each of the tasks until the maximum number of iterations is reached. No special consideration is made for the optimizer or weighing of objectives. The results show that intermediate-multi-task training improves performance over the baseline for RTE, but this improvement is less than when only MNLI is used for intermediate-task training. Though not desirable if RTE is the only target task, such intermediate-multi-task training yields a better multi-task model that performs well on both datasets: the joint (single) model achieved 75.0 on RTE and 86.2 on MNLI, both of which are better than their single-task baselines. In some cases, the increased performance for one task (MNLI) might be preferable to that on another (RTE). We note that this observation is similar to the one of Phang et al. (2018).

**SpEx-BERT on STILTs is more robust than BERT on STILTs when training data is limited.** In Table 3b, we present results for the same models (BERT and SpEx-BERT) being fine-tuned with sub-sampled versions of the dataset. For this experiment, we follow Phang et al. (2018) and subsample 1000 data points at random without replacement and choose the best development set accuracy across several random restarts. The rest of the experimental setup remains unchanged. When used in conjunction with STILTs, the performance improves as expected and, in a majority of cases, significantly exceeds that of the corresponding baseline that does not use span-extraction.

## 6 DISCUSSION

### 6.1 PHRASING THE QUESTION

As described in Section 3, when converting any of the classification or regression problems into a span-extraction one, the possible classes or bucketed values need to be presented in natural language as part of the input text. This leaves room for experimentation. We found that separation of naturally delimited parts of the input text into source and auxiliary text was crucial for best performance. Recall that for question answering, the natural delimitation is to assign the given context document as the source text and the question as the auxiliary text. This allows the span-decoder to extract a span from the context document, as expected. For single-sentence problems, there is no need for delimitation and the correct span is typically not found in the given sentence, so it is treated as auxiliary text.

Natural language descriptions of the classes or allowable regression values are provided as source text for span extraction. For two-sentence problems, the natural delimitation suggests treating one sentence as source text and the other as auxiliary. The classification or regression choices must be in the source text, but it was also the case that one of the sentences must also be in the source text. Simply concatenating both sentences and assigning them as the source text was detrimental for tasks like MNLI.

For the case of classification, when experimenting with various levels of brevity, we found that simpler is better. Being terse eases training since the softmax operation over possible start and end locations is over a relatively smaller window. While more detailed explanations might elaborate on what the classes mean or otherwise provide additional context for the classes, these potential benefits were outstripped by increasing the length of the source text. We present these results on the development set of the MNLI dataset with $BERT_{BASE}$ in Table 4b. For regression, there exists a trade-off between brevity and granularity of the regression. We found that dividing the range into $10 - 20$ buckets did not appreciably change the resulting correlation score for STS-B.

## 6.2 A FULLY JOINT MODEL WITHOUT TASK-SPECIFIC PARAMETERS

Unlike similar approaches using task-specific heads Liu et al. (2019a), SpEx-BERT allows for a single model across a broader set of tasks. This makes possible a single, joint model with all parameters shared. We present the results of this experiment in Table 5 in the Appendix; we multi-task over all datasets considered so far. Multi-task performance exceeds single-task performance for many of the question answering datasets (ZRE, SRL, CQA) as well as the classification dataset RTE. In some cases, these improvements are drastic (over $9\%$ accuracy). Unfortunately, the opposite is true for the two tasks that are the greatest source of transfer, MNLI and SQuAD, and the remaining GLUE tasks. Understanding why such vampiric relationships amongst datasets manifest, why any particular dataset appears beneficial, neutral, or detrimental to the performance of others, and why question answering tasks appear more amenable to the fully-joint setting remain open questions. Nonetheless, a purely span-extractive approach has allowed us to observe such relationships more directly than in settings that use multiple task-specific heads or fine-tune separately on each task. Because some tasks benefit and others suffer, these results present a trade-off. Depending on which tasks and datasets are more pertinent, multi-task learning might be the right choice, especially given the ease of deploying a single architecture that does not require any task-specific modifications.

Joint models for NLP have already been studied Collobert et al. (2011); McCann et al. (2018); Radford et al. (2019) with a broad set of tasks that may require text generation and more general architectures. These approaches have yet to perform as well as task-specific models on common benchmarks, but they have demonstrated that large amounts of unsupervised data, curriculum learning, and task sampling strategies can help mitigate the negative influence multitasking tends to have on datasets that are especially good for transfer learning. This work represents a connection between those works and work that focuses on task-specific fine-tuning of pre-trained architectures.

## 7 CONCLUSION

With the successful training of supervised and unsupervised systems that rely on increasingly large amounts of data, more of the natural variation in language is captured during pre-training. This suggests that less inductive bias in the design of task-specific architectures might be required when approaching NLP tasks. We have proposed that the inductive bias that motivates the use task-specific layers is no longer necessary. Instead, a span-extractive approach, common to question answering, should be extended to text classification and regression problems as well. Experiments comparing the traditional approach with BERT to SpEx-BERT have shown that the span-extractive approach often yields stronger performance as measured by scores on the GLUE benchmark. This reduces the need for architectural modifications across datasets or tasks, and opens ways for applying methods like STILTs to question answering or a combination of text classification, regression, and question answering datasets to further improve performance. Experiments have further shown that span-extraction proves more robust in the presence of limited training data. We hope that these findings will promote further exploration into the design of unified architectures for a broader set of tasks.

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

# A  MULTITASKING RESULTS

Below is the table that supports the commentary in Section 6.2.

| | SST | MRPC | QQP | MNLI | RTE | QNLI | CoLA | STS | SQuAD | ZRE | SRL | CQA |
|---|---|---|---|---|---|---|---|---|---|---|---|---|
| Individual Models | | | | | | | | | | | | |
| BERT$_{LARGE}$ | 92.5 | **89.0** | **91.5** | 86.2 | 70.0 | **92.3** | 62.1 | **90.9** | **84.0** | 69.1 | 90.3 | 60.3 |
| SpEx-BERT$_{LARGE}$ | **93.7** | 88.9 | 91.0 | **86.3** | 69.8 | 91.8 | **64.8** | 89.5 | **84.0** | 69.1 | 90.3 | 60.3 |
| Multi-task Models (best joint single model) | | | | | | | | | | | | |
| SpEx-BERT$_{LARGE}$ | 92.4 | **87.5** | 90.9 | 85.0 | 71.1 | **91.3** | **58.8** | 89.2 | 80.4 | 75.0 | 97.7 | 61.0 |
| →MNLI | **93.2** | 87.0 | 90.9 | **85.6** | 81.2 | **91.3** | 57.9 | 90.1 | 80.5 | 76.6 | 97.7 | 61.5 |
| →SQuAD | 92.2 | 87.0 | **91.0** | 85.3 | 80.9 | 91.2 | 52.0 | 90.1 | **80.6** | **78.8** | 97.7 | **63.4** |
| →MNLI→SQuAD | 92.3 | 90.9 | 90.8 | 85.2 | **84.1** | 90.9 | 52.1 | **90.2** | **80.6** | 75.3 | **97.8** | 61.5 |
| Multi-task Models (maximum individual score for each dataset during the course of training) | | | | | | | | | | | | |
| SpEx-BERT$_{LARGE}$ | 93.0 | 88.5 | 91.0 | 85.2 | 73.3 | 91.4 | 59.8 | 88.9 | 81.9 | 77.8 | 97.7 | 64.7 |
| →MNLI | **93.2** | 89.7 | 90.8 | **85.7** | 84.1 | **91.6** | **59.9** | 89.8 | 81.4 | 78.2 | 97.7 | 63.3 |
| →SQuAD | 92.9 | 89.2 | **91.1** | 85.4 | 84.1 | 91.4 | 56.1 | 90.1 | 82.8 | **79.6** | **97.8** | **65.3** |
| →MNLI→SQuAD | 92.7 | **91.4** | 90.8 | 85.4 | **85.2** | 91.2 | 57.5 | **90.2** | **83.2** | 77.5 | **97.8** | 64.8 |

Table 5: Development set performance metrics on a single (joint) model obtained by multi-tasking on all included datasets. We include best single-task performances (without STILTs), labeled as individual models, for the sake of easier comparison. We divide the remaining into two parts – in the first, the scores indicate the performance on a single snapshot during training and not individual maximum scores across the training trajectory. In the second, we include the best score for every dataset through the training; note that this involves inference on multiple model snapshots. For the models trained with STILTs, the SpEx-BERT model is first fine-tuned on the intermediate task by itself after which the model is trained in multi-tasking fashion. **Bold** implies best in each column (i.e., task).

