# OpenReview forum: "Unifying Question Answering, Text Classification, and Regression via Span Extraction"
_ICLR.cc/2020/Conference — Reject_

### Official Review · AnonReviewer1 · 2019-10-15
**Official Blind Review #1**

**Rating:** 3

**Review:**

This paper asks whether it works to remove task-specific heads and treat classification and regression problems as span extraction, by formatting problems in such a way that a single span extraction model can be used.  This is a reasonable question to ask, and the authors performed a very large number of experiments attempting to answer this question.  The authors claim that using span extractive models instead of task-specific heads yields improved performance over separate heads.

My main concern with this work is actually with something that would otherwise be a strength - the very large number of experiments.  Looking at the results tables, I come to a different conclusion from the authors: there does not appear to be a significant difference between using a single head or using multiple heads (this is still an interesting conclusion).  The numbers presented all appear to be within the bounds of random fluctuations, which are not controlled for at all with standard statistical testing methodologies.  And with the very large number of experiments, there are bound to be some that stand out.  This is especially true given the methodology used for small datasets - even if it was used by prior work, it is still not statistically sound to take the max of 20 samples from a training distribution and report the difference without any other statistical testing.

As an example, look at table 3b.  This is claimed as a big win for SpExBERT when comparing the MNLI fine-tuned version.  But if you look at the other rows in the table, SpExBERT is worse than BERT by a *larger margin* than it is better in the MNLI case (contradicting the claimed result from table 2).  And this general trend is seen across the tables - the differences are small and inconsistent, making it look very much like we are just seeing random fluctuations.  This must be controlled for statistically in order to make any valid conclusions.  The one possible exception here seems to be SST.  Those results on the dev set do indeed seem to be more consistent, which is interesting, and hints at the utility of using "positive" and "negative" as natural language descriptions, as the authors claim.  It's not very convincing, however, as the test set difference is very small, and SpExBERT had more opportunities to find a good dev set number, as it had more experiments.

My second major concern is with the experimental set up.  The authors want to claim that using a unified span extraction format yields superior performance to having separate heads.  But there are baselines missing to really demonstrate this claim.  The multiple head setup isn't really evaluated as a baseline in most of the experiments (e.g., using SQuAD / other QA datasets as an intermediate task in table 2, using BERT for any of the QA datasets in table 3).  So, even if the above issue of statistical testing were solved, it would still be very hard to evaluate the claim, as the proper comparisons are not present in the majority of cases.

My main conclusion from reading this paper is that it does not appear to matter what head you use for these particular datasets.  This is an interesting result, though it is not the one that is claimed in the paper.  I think it would be very challenging to extend this approach to a broader set of tasks, however, as the authors suggest towards the end of the paper.  How do the authors propose handling cases where it is not feasible to put all possible outputs in the context?  This includes any generative task (including generative QA) and any kind of structured output (like parsing, tagging, etc.), or a regression task that does not lend itself well to bucketing.

Minor issues:

I would be careful about claiming that you've successfully captured regression in a span extraction format.  You have one regression task, and it's one where the original labels were already bucketed, so the bucketing makes sense.  I am very skeptical that this would actually work for more general regression.

Re the paragraph titled "SpEx-BERT improves on STILTs": Note that SpExBERT requires additional data preprocessing and hand-written templates when switching between tasks, which is not necessary in the method of Phang et al.  There are always tradeoffs.  Neither using separate heads nor doing your additional processing are very hard, so I wouldn't make as big a deal out of this as you do.  If you want to talk about one of the methods using more "human inductive bias", or whatever, the hand-written templates in yours might actually be _more_ "incidental supervision" than a very small difference in model architecture.  But again, the difference here is very small, not worth noting in the paper.

**Experience Assessment:**

I have published in this field for several years.

**Review Assessment: Checking Correctness Of Derivations And Theory:**

I carefully checked the derivations and theory.

**Review Assessment: Checking Correctness Of Experiments:**

I carefully checked the experiments.

**Review Assessment: Thoroughness In Paper Reading:**

I read the paper thoroughly.

---

> ### Author Response · Authors · 2019-11-13
> **Rebuttal for R1**
>
> Thank for your careful review. Your commentary on the experimentation is spot-on, we are grateful for such detailed comments. We address your concerns below.
>
>
> > My main concern with this work is actually with something that would otherwise be a strength - the very large number of experiments.  Looking at the results tables, I come to a different conclusion from the authors: there does not appear to be a significant difference between using a single head or using multiple heads (this is still an interesting conclusion)...
>
> As we also state in our rebuttals for R2/R3, our motivation was not to claim superiority of the unified approach but to emphasize that it performs comparably both in the vanilla BERT setting and when combined with heuristics such as STILTs. This eschewing of task-specific heads/losses is what we wished to highlight. In our list of contributions (Section 1.1), we were careful not to underscore any performance improvements on the GLUE tasks or claim superiority of the approach; we believe, and as you probably would agree, the paper stands with solely those as contributions. However, as you pointed out, we inadvertently did introduce verbiage through parts of the paper that confuses the reader about our claims/motivation and will rewrite those in an updated manuscript.
>
>
> > My second major concern is with the experimental set up.  The authors want to claim that using a unified span extraction format yields superior performance to having separate heads.  But there are baselines missing to really demonstrate this claim.  The multiple head setup isn't really evaluated as a baseline in most of the experiments (e.g., using SQuAD / other QA datasets as an intermediate task in table 2, using BERT for any of the QA datasets in table 3).  So, even if the above issue of statistical testing were solved, it would still be very hard to evaluate the claim, as the proper comparisons are not present in the majority of cases…
>
> In our experiments, QA tasks did not out-perform MNLI as intermediate tasks (with the exception of QNLI; which had to be excluded given overlap with SQuAD) so we did not choose them for STILT-ing on for the baseline. As above, the goal of experimenting with QA/classification tasks together was to show the seamlessness of the process given the unification and less so to claim improvements. We briefly allude to this intention in Section 5 and in the conclusion (“...opens ways for applying methods like STILTs to question answering or a combination of text classification, regression, and question answering datasets ...”) but will clarify it in the updated manuscript.
>
>
> > I would be careful about claiming that you've successfully captured regression in a span extraction format.  You have one regression task, and it's one where the original labels were already bucketed, so the bucketing makes sense.
>
> The original labels for STS-B are actually floating point values between 0 and 5 but we agree with your comment on generalizing this to other regression problems. We will add more commentary around this.
>
> > the paragraph titled "SpEx-BERT improves on STILTs": Note that SpExBERT requires additional data preprocessing and hand-written templates when switching between tasks, which is not necessary in the method of Phang et al.  There are always tradeoffs.
>
> Although the processing/templating is rather straightforward, your comment is true. Thanks for pointing it out; we will explicitly add commentary around the tradeoffs.

---

### Official Review · AnonReviewer3 · 2019-10-23
**Official Blind Review #3**

**Rating:** 3

**Review:**

This submission proposes a span-extraction approach to unify QA, text classification and regression tasks. The extensive empirical studies presented in this paper demonstrate that the proposed method can improve the performance by a small margin across multiple datasets and tasks. Overall, I find that the idea of unifying QA, text classification and regression is interesting by itself, but the experiments cannot justify their claims well mainly due to the mixed results.

I have the following concerns:

0. Compared to decalNLP (https://github.com/salesforce/decaNLP), this new approach seems unable to handle as many types of tasks as decalNLP. It is not clear to me what is the main advantage. As discussed in the Related Work Section, decalNLP needs to fix the classifier a-priori, but this submission's method needs a natural language description, which seems more difficult to implement in practice.
1. The results are mixed. For example, SpEx-BERT underperforms BERT on RTE and QNLI in Table 2 and Span-extractive multi-task learning results in weaker single-task models as shown in Table 4a. Furthermore, an error analysis would be helpful here.

**Experience Assessment:**

I have read many papers in this area.

**Review Assessment: Checking Correctness Of Derivations And Theory:**

N/A

**Review Assessment: Checking Correctness Of Experiments:**

I assessed the sensibility of the experiments.

**Review Assessment: Thoroughness In Paper Reading:**

I read the paper at least twice and used my best judgement in assessing the paper.

---

> ### Author Response · Authors · 2019-11-13
> **Rebuttal for R3**
>
> Thank you for your review.
>
> > Compared to decalNLP (https://github.com/salesforce/decaNLP), this new approach seems unable to handle as many types of tasks as decalNLP. It is not clear to me what is the main advantage. As discussed in the Related Work Section, decalNLP needs to fix the classifier a-priori, but this submission's method needs a natural language description, which seems more difficult to implement in practice.
>
> We see the decaNLP paper as comprising three contributions: (a) a framework that unifies NLP via question answering, (b) the underlying challenge with datasets and metrics, and (c) the MQAN model. While MQAN/decaNLP enabled the unification of various NLP tasks (including sequence generation), the performance on each of the individual tasks suffered; at times the individual tasks were 15 nF1 lower than the state-of-the-art at the time, and even more now. In our work, we wished to prioritize maintaining performance on individual tasks while still unifying as large a class of NLP tasks as possible. Indeed, our performance is comparable (if not better) than the BERT baseline. For this purpose, we took inspiration from the decaNLP setup of posing tasks in the same input format but differ from it in all other ways. Finally, our submission does not need a natural language description. Just as decaNLP, we simply list the available choices of classification in our input (see Fig. 1 of our paper, for instance).
>
>
> > The results are mixed. For example, SpEx-BERT underperforms BERT on RTE and QNLI in Table 2 and Span-extractive multi-task learning results in weaker single-task models as shown in Table 4a. Furthermore, an error analysis would be helpful here.
>
> We apologize for the misunderstanding. As we also state in our rebuttals for R1/R2, our motivation was not to claim superiority of the unified approach but to emphasize that it performs comparably both in the vanilla BERT setting and when combined with heuristics such as STILTs. This eschewing of task-specific heads/losses is what we wished to highlight. In our list of contributions (Section 1.1), we were careful not to underscore any performance improvements on the GLUE tasks or claim superiority of the approach; we believe the paper stands with solely those as contributions. It is true that in some cases, BERT is better, however, in aggregate the unified model stays competitive both in terms of the absolute score on the vanilla model and improvements due to heuristics such as STILTs. Through verbiage in some sub-sections, we inadvertently confused the reader about our claims/motivation and will rewrite those in an updated manuscript. We will also add an error analysis with attention visualizations to the manuscript.

---

### Official Review · AnonReviewer2 · 2019-10-23
**Official Blind Review #2**

**Rating:** 3

**Review:**

This paper introduces a method for converting sentence pair classification tasks and sentence regression tasks into span into span extraction tasks, by listing all the possible classes (entailment, contradiction, neural) or the discretized scores (0.0, 0.25 ...) and concatenating them with the source text. With this formulation, one can train a BERT-based span-extraction model (SpEx-BERT) on classification, regression, and QA tasks without introducing any task-specific parameters. The purposed SpEx-BERT model achieves moderate improvement (0.3 points) over the BERT-large baseline on the GLUE test set when fine-tuned on intermediate STILTs tasks (Phang et al., 2018).

Strengths:
- Extensive finetuning/intermediate-finetuning experiments on a range of NLP tasks.
- The paper is mostly well-written and easy to follow.

Weaknesses:
- This paper presents a lot of experiments. But it seems that the most useful / head-to-head comparison against the BERT model are the last 2 rows in Table 2 with the GLEU results, where the improvement is moderate.
- The idea of expressing various NLP tasks (including textual entailment and text classification) as question-answer has been well-explored in decaNLP (McCann et al., 2018). It would be nice if the authors could elaborate more on how the proposed method differs from theirs.

Other comments/suggestions:
- Likely typo in abstract: "fixed-class, classification layers for text classification"


**Experience Assessment:**

I have published one or two papers in this area.

**Review Assessment: Checking Correctness Of Derivations And Theory:**

N/A

**Review Assessment: Checking Correctness Of Experiments:**

I assessed the sensibility of the experiments.

**Review Assessment: Thoroughness In Paper Reading:**

I read the paper at least twice and used my best judgement in assessing the paper.

---

> ### Author Response · Authors · 2019-11-13
> **Rebuttal for R2**
>
> Thank you for your review. We are glad that you found our paper well-written. We address some of your concerns next.
>
> > This paper presents a lot of experiments. But it seems that the most useful / head-to-head comparison against the BERT model are the last 2 rows in Table 2 with the GLEU results, where the improvement is moderate.
>
> We apologize for the misunderstanding. As we also state in our rebuttals for R1/R3, our motivation was not to claim superiority of the unified approach but to emphasize that it performs comparably both in the vanilla BERT setting and when combined with heuristics such as STILTs. This eschewing of task-specific heads/losses is what we wished to highlight. In our list of contributions (Section 1.1), we were careful not to underscore any performance improvements on the GLUE tasks or claim superiority of the approach; we believe the paper stands with solely those as contributions. Through verbiage in some sub-sections, we inadvertently confused the reader about our claims/motivation and will rewrite those in an updated manuscript.
>
> > The idea of expressing various NLP tasks (including textual entailment and text classification) as question-answer has been well-explored in decaNLP (McCann et al., 2018). It would be nice if the authors could elaborate more on how the proposed method differs from theirs.
>
> We see the decaNLP paper as comprising three contributions: (a) a framework that unifies NLP via question answering, (b) the underlying challenge with datasets and metrics, and (c) the MQAN model. While MQAN/decaNLP enabled the unification of various NLP tasks (including sequence generation), the performance on each of the individual tasks suffered; at times the individual tasks were 15 nF1 lower than the state-of-the-art at the time, and even more now. In our work, we wished to prioritize maintaining performance on individual tasks while still unifying as large a class of NLP tasks as possible. Indeed, our performance is comparable (if not better) than the BERT baseline. For this purpose, we took inspiration from the decaNLP setup of posing tasks in the same input format but differ from it in all other ways.

---

### Decision · Program_Chairs · 2019-12-19

**Decision:**

Reject

**Comment:**

(I acknowledge reading authors' recent note on decaNLP.)

This paper proposes a span extraction approach (SpExBERT) to unify question answering, text classification and regression. Paper includes a significant number of experiments (including low-resource and multi-tasking experiments) on multiple benchmarks. The reviewers are concerned about lack of support on author's claims from the experimental results due to seemingly insignificant improvements and lack of analysis regarding the results. Hence, I suggest rejecting the paper.